# Oak-YOLO: A high-performance detection model for automated Oak seed defect identification

**Hao Li**[1], **Zhuqi Li**[1], **Dongkui Chen**[1], **Wangyu Wu**[2], **Xuanlong He**[1], **Hongbo Mu**[1]*

**1** College of Science, Northeast Forestry University, Harbin, Heilongjiang, China, **2** University of Liverpool, Liverpool, United Kingdom

* mhb506@nefu.edu.cn

**Data availability statement:** The dataset analyzed during the current study are available in the Figshare repository at DOI 10.6084/m9.figshare.29072015 (https://doi.org/10.6084/m9.figshare.29072015).

## Abstract

Oak seeds are highly susceptible to pest infestations due to their elevated starch content, which significantly impairs germination and subsequent growth. To address this challenge, we developed a high-resolution imaging system and proposed an improved YOLO-based model named Oak-YOLO for efficient and accurate defect detection in oak seeds. The proposed model enhances the YOLOv8 architecture by incorporating EfficientViT as the backbone to improve global feature extraction, and integrates a Ghost-DynamicConv detection head to enhance the representation of small and irregular defects such as insect holes and cracks. Additionally, the WIoUv3 loss function is introduced to optimize bounding box regression for complex target shapes and overlapping instances.Extensive experiments were conducted on both single-object and multi-object datasets. Oak-YOLO achieved a mAP50 of 94.5%, an F1-score of 95.3%, and a precision of 94.% on the oak-intensive dataset, with an inference speed of 132.2 FPS. Cross-device validation using mobile-captured images further demonstrated the model's robustness, achieving mAP50 scores of 94.7% and 93.8% on different smartphone test sets. Comparative evaluations show that Oak-YOLO outperforms existing YOLO models, including YOLOv9 to YOLOv12, by delivering a favorable trade-off between detection accuracy and computational efficiency. These results highlight the potential of Oak-YOLO as a practical solution for real-time seed quality inspection in forestry applications.

## Introduction

In modern forestry production, oak seeds, due to their high starch content, are particularly vulnerable to insect infestations [1]. These infestations not only affect the quality of the seeds but also significantly reduce their germination rate and growth potential. To address these issues, chemical pesticides have traditionally been widely used. However, the effectiveness of this approach has been limited, with insect damage remaining prevalent [2], and the overuse of chemical pesticides potentially causing adverse environmental effects.

**Funding:** The Fundamental Research Funds for the Central Universities-No. 2572023DJ02. The funders provided support in reviewing and editing the manuscript.

**Competing interests:** The authors have declared that no competing interests exist.

Traditional methods for screening oak seeds primarily include the weighing method and the water immersion method. The weighing method estimates seed damage by comparing the seed's weight, while the water immersion method relies on the buoyancy difference of seeds in water to distinguish healthy seeds. However, these methods have significant limitations, as they cannot accurately detect subtle internal insect damage, leading to suboptimal screening results and affecting overall seed quality and planting success [3].

Given the limitations of traditional methods in detecting defects in oak seeds, this study introduces machine vision technology for efficient and accurate identification of oak seed defects. Traditional machine vision methods rely on manually designed features such as color, shape, texture, and spectral information. These methods have been widely applied in plant seed detection and classification. For example, Wang et al. [4] proposed a maize seed recognition method based on genetic algorithms (GA) and multi-class support vector machines (SVM), While Nguyen-Quoc et al. [5] utilized image preprocessing,HOG descriptor, and various imputation techniques combined with SVM classifiers to classify different rice seeds.

In contrast to traditional methods, deep learning techniques can automatically extract rich, multi-level features from images, enabling higher detection accuracy and efficiency in complex environments. YOLO frameworks [6–8], along with region-based models like Faster R-CNN [9] and single-shot detectors like SSD [10], enable fast and accurate defect identification in real-time. Recent advancements, such as Swin Transformer-based models [11] and transfer learning approaches [12], have further enhanced feature extraction and adaptability. Standard datasets like Pascal VOC [13] and COCO [14] provide benchmarks for evaluating these models. These innovations streamline seed quality assessment, ensuring higher efficiency and accuracy for agricultural applications.For instance, Mukasa et al. [15] used DD-SIMCA, SVM, and deep learning classifiers to distinguish between triploid and diploid watermelon seeds; Kurtulmus [16] proposed a sunflower seed classification method based on deep convolutional neural networks (CNNs), enhancing classification performance through various CNN architectures; Wang et al. [17] combined hyperspectral imaging with deep learning, proposing a novel CNN-LSTM model for maize seed variety identification that showed excellent performance; Shi et al. [18] employed iPhone images and deep learning methods, using data augmentation and transfer learning strategies to improve barley seed variety identification accuracy. Barrio-Conde et al. [19] successfully classified high oleic sunflower seed varieties using deep learning algorithms, while Bi et al. [20] proposed a maize seed recognition method based on an improved Swin Transformer, which incorporated feature attention mechanisms and multi-scale feature fusion to enhance recognition accuracy; Thakur et al. [21] introduced deep transfer learning photon sensors, improving seed quality evaluation accuracy and efficiency through laser backscattering and deep learning.deep learning technology has significantly advanced the development of crop seed detection. However, existing computer vision research is primarily focused on the detection of seeds from other crops, while the detection of tree seeds has received relatively less attention.

In this study, we constructed a dedicated dataset for oak seed defect detection. The dataset was meticulously curated to cover various defect types, and incorporated diverse lighting conditions, backgrounds, and spatial arrangements. To further enhance the dataset's robustness, we employed automatic data augmentation techniques, including rotation, brightness adjustment, and background variation, ensuring a wide range of sample diversity.Building on this comprehensive dataset, we then proposed targeted improvements specifically aimed at enhancing small target detection in oak seed defect analysis. To improve the YOLO model's performance in identifying small defects, we introduced a multi-scale feature fusion mechanism designed to preserve high-resolution feature information. This approach ensures that

small defects, such as cracks and insect holes, are effectively retained within high-level feature maps.

Additionally, we incorporated an improved attention mechanism to enhance boundary separation between adjacent seeds, reducing interference caused by overlapping targets. To address the challenge of detecting irregularly shaped defects, we proposed a novel loss function that enables the model to accurately fit the boundaries of cracks and insect holes using rotated rectangular boxes.

Experimental results demonstrate that the proposed detection framework significantly outperforms traditional methods in detecting oak seed defects. The model effectively addresses the limitations of the YOLO architecture, particularly in small target detection, overlapping object handling, irregular shape adaptation, and data labeling challenges. This comprehensive improvement greatly enhances the accuracy and efficiency of oak seed quality inspection, making the framework highly suitable for practical applications in forestry production.

## Materials and methods

### Selection of seed

The oak seeds used in this study were purchased from oak farmers in Xinxu Town, Suqian, Jiangsu, China. The seeds were cleaned to remove plant debris, soil, dust, and stones. Subsequently, the seeds were stored at temperatures ranging from -20°C to -30°C for two weeks to eliminate residual pests.

### Photographic equipment

As shown in Fig 1.The image acquisition system utilized a Canon camera (ME2P-G-P, Hangzhou, China) with a resolution of 4508×4096. An adjustable lighting setup was employed to facilitate image capture, with the light source set to a color temperature of 5000K to replicate natural lighting conditions. A circular LED light was positioned at a 45° angle, with the illumination diffused through white fabric to ensure soft, uniform lighting and to minimize specular reflections that could impede the detection of subtle defects. Based on measurements, the optimal focusing distance between the camera and the seeds was maintained between 12 and 15 cm. The platform was designed to enable flexible adjustments of shooting angles and heights, allowing for comprehensive capture of seed surface features from

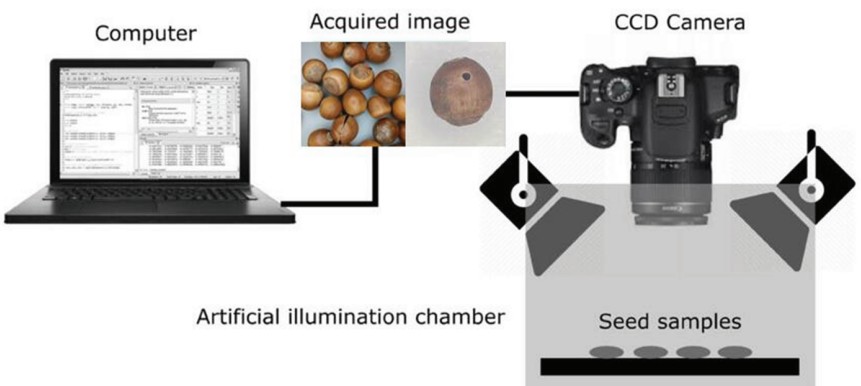

**Fig 1. The process of image acquisition.**

various perspectives. To maintain consistent image exposure, the ISO setting of all imaging devices was fixed at 800.

## Segmentation of seed images

In this study, we employed the automated segmentation algorithm EfficientSAM to annotate the dataset. As shown at Fig 2. The original images were first denoised using median filtering to remove background noise and interference on the seed surface. Subsequently, the images were converted to the HSV color space to enhance the contrast between the seeds and the background. In the HSV space, histogram equalization was applied to the brightness channel to enhance seed surface details:

$$V'(x,y) = \frac{L-1}{M \cdot N} \sum_{i=0}^{V(x,y)} h(i) \tag{1}$$

$L$ denotes the maximum grayscale level, $M \times N$ is the total number of pixels, and $h(i)$ represents the frequency of the grayscale level $i$. This equalization further improves the detail of the seed surface.we applied an adaptive threshold segmentation method to separate the seed region from the background, generating an initial binary mask:

$$T(x,y) = \frac{1}{|N(x,y)|} \sum_{(i,j)\in N(x,y)} (I(i,j) - C) \tag{2}$$

In this equation, $T(x,y)$ denotes the threshold value, $N(x,y)$ is the neighborhood around the pixel, and $C$ is a constant for adjustment. This formula calculates the average value within the local neighborhood and adjusts it to achieve binarization. Subsequently, morphological operations, including dilation and erosion, were employed to refine the mask edges. Dilation

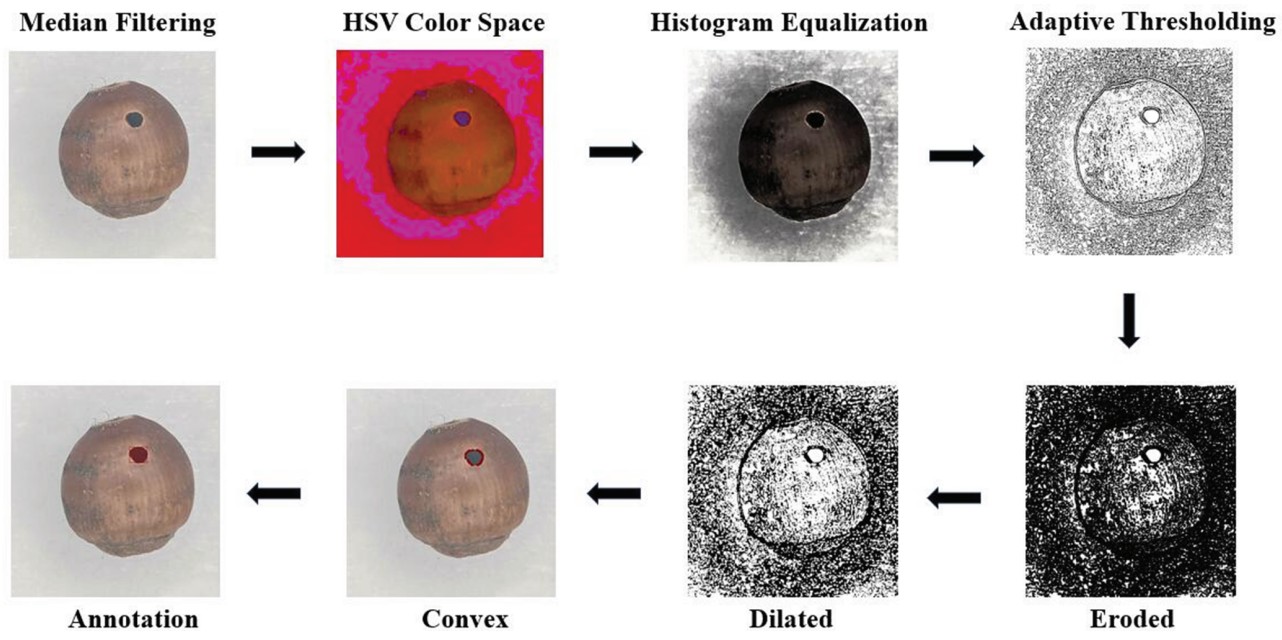

**Fig 2. Segmentation process.**

helps connect adjacent seed regions, while erosion removes noise, making seed contours clearer. After these processing steps, the resulting binary mask accurately represents the seed area.

To annotate irregular defect areas such as cracks and insect holes, we introduced the convex hull algorithm. Traditional rectangular annotations often fail to capture complex defect boundaries, especially for irregular geometries like cracks and insect holes. The convex hull method effectively addresses this limitation by precisely enclosing defect areas.

Given a set of points $P = \{P_1, P_2, \ldots, P_n\}$ representing the boundary of the defect region, the convex hull $CH(P)$ is the smallest convex polygon that encloses the set $P$. Assuming $P_i$, $P_j$, and $P_k$ are three points in the set, the cross product of vectors $P_iP_j$ and $P_iP_k$ is calculated as follows:

$$\text{cross}(P_i, P_j, P_i, P_k) = (x_j - x_i) \cdot (y_k - y_i) - (y_j - y_i) \cdot (x_k - x_i) \qquad (3)$$

After annotation, to match the YOLO object detection model's input format, all annotated bounding box information was converted to the standard YOLO label format. Each label contains five values: the object class, the center coordinates of the bounding box $(x,y)$, the width $(w)$, and the height $(h)$. These values were normalized, and the segmented data were further augmented to increase model robustness. The augmented data covers diverse lighting conditions, angle variations, and noise simulation to match real-world production environments.

To evaluate the effectiveness of the EfficientSAM segmentation algorithm, we used a subset of manually annotated seed images as a control group for accuracy comparison. Compared with traditional rectangular annotation methods, EfficientSAM significantly improved segmentation accuracy, especially for complex defect shapes. The convex hull annotation method provided more precise coverage of defect areas. The comparison results are presented in Table 1.

## Data augmentation

The color images were first converted to grayscale, reducing both storage space and computational complexity by approximately one-third. Subsequently, each pixel value was multiplied by a randomly generated brightness factor ranging from 0.5 to 1.5, enhancing the model's robustness to variations in illumination. Next, the images were randomly rotated within an angle range of –90° to 90°, and Gaussian noise was added with a 50% probability, using a noise matrix generated from a Gaussian distribution with a mean of 0 and a standard deviation of 0.01. Finally, multiple augmented images were randomly combined into a single large image using 2×2, 3×3, or 4×4 grid patterns, and the corresponding labels were updated based on the offset caused by the image stitching. Fig 4 illustrates the augmentation process across different samples.

Subsequently, Gaussian noise was added to the images with a 50% probability. The noise was generated from a Gaussian distribution with a mean of 0 and a standard deviation of 0.01, and was randomly superimposed on the pixel values of the images. To further enhance the

**Table 1. Comparison of annotation methods.**

| Annotation Method | Model | Precision | Recall | mAP@0.5 | mAP@0.5:0.95 |
|---|---|---|---|---|---|
| Conventional | YOLOv8n | 0.794 | 0.817 | 0.783 | 0.527 |
| EfficientSAM | YOLOv8n | 0.853 | 0.861 | 0.891 | 0.679 |

dataset, multiple images were concatenated into a single large image, with random grid sizes (2x2, 3x3, or 4x4), and corresponding label files were generated. After each concatenation, the coordinates of the labels were adjusted according to the offset of the concatenated images.

Fig 3(a) illustrates the distribution of *x* and *y* coordinates for all annotations in the -seed dataset. The figure shows that annotation centers are evenly distributed across the image, with no significant clustering or outliers. This indicates that the seed positions exhibit good diversity, which helps the model learn a wider range of features. Additionally, the distributions of annotation width (*w*) and height (*h*) are shown, with *w* and *h* primarily concentrated between 0.01 and 0.05. This suggests a certain degree of clustering in annotation sizes, likely due to denser seed placement in specific image regions, leading to more similar annotation sizes in those areas. In contrast, the distribution of *x* and *y* coordinates in Fig 3(b) appears more concentrated, likely reflecting the characteristics of densely packed detection samples.

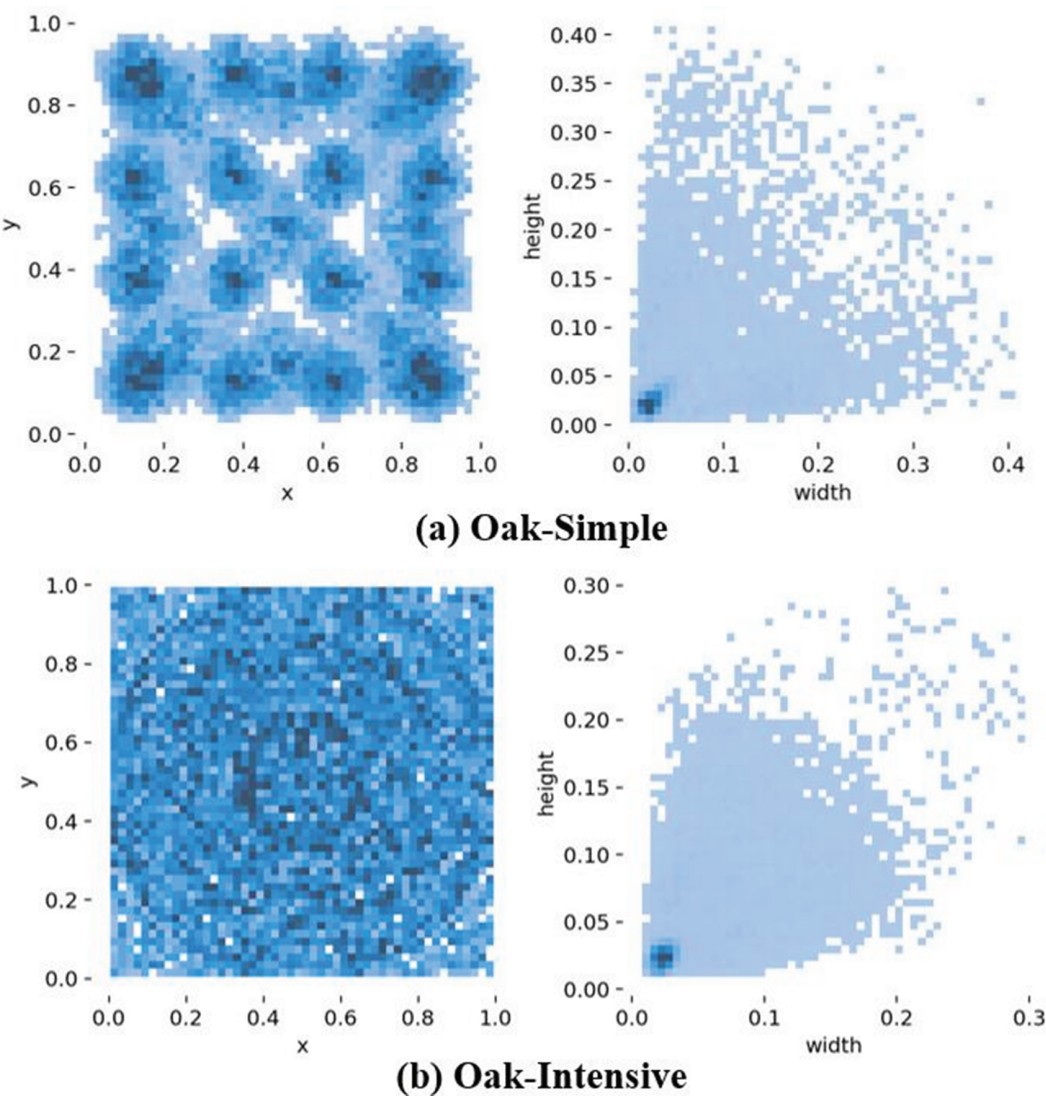

**Fig 3. Comparison of concat-image and densely detected data.** (a) Features of concat-image data. (b) Features of densely detected data.

Similarly, *w* and *h* are concentrated between 0.015 and 0.04, further highlighting the annotation patterns in densely populated regions.

After applying the aforementioned augmentation techniques, the number of single-object seed images in the dataset was increased from 2,000 to 2,537, while the multi-object dataset grew from 1,572 to 1,949 images. The dataset was split into training, validation, and test sets using an 8:1:1 ratio with stratified random sampling to ensure that the class distribution remained consistent across all subsets.Specifically, the training set consists of 4,538 images, while the testing and validation sets each contain 567 images. The test set was exclusively reserved for final performance evaluation.The single-object dataset is named oak-simple, and the multi-object dataset is named oak-intensive, as summarized in Table 2.

## Oak-YOLO

YOLOv8 [22] exhibits high real-time detection capability and accuracy optimization, making it suitable for object detection tasks. However, YOLOv8 still has limitations in detecting irregular-shaped defects in oak seeds, as the rectangular bounding box cannot accurately describe the edges, leading to incomplete detection or boundary deviation. Additionally, its real-time performance and efficiency are limited when deployed on resource-constrained embedded devices.

As shown in Fig 4, we improved YOLOv8 in three aspects: 1) Introducing the Ghost-Dynamic prediction head, which combines shallow and deep features to effectively enhance the detection accuracy of small defects such as cracks and wormholes; 2) Upgrading the YOLOv8 backbone by replacing traditional CNN modules with EfficientViT to improve the capture of global features and long-range dependencies; 3) Employing the WIoUv3 loss function to optimize IoU calculation for small and overlapping defects, ensuring that the predicted bounding box shape better matches the defect characteristics.

**Ghost-dynamic.** In the YOLOv8 model, instead of using the shallow feature map $F_2$ with limited semantic information, the deeper feature maps $F_3$, $F_4$, and $F_5$, extracted from the backbone network are passed into the neck for feature fusion. These feature maps, after undergoing multiple convolutional down-sampling layers, gradually expand their receptive fields. The deeper feature maps contain more rich semantic information, sufficient to handle typical object detection tasks. However, in the case of oak seed defect detection, particularly for small and complex defects such as insect holes and cracks, using deeper features may lead to information dilution, thereby reducing both detection precision and localization accuracy. This is because these defects often appear as small objects, and the positional information in deeper features is relatively sparse and blurred due to the convolutional layers, making precise defect localization and classification more challenging [23]. Moreover, due to insufficient information extracted by the prediction head from the feature maps, the detection accuracy of defects is compromised. Similar, overlapping, and partially occluded defects further exacerbate the difficulty in object detection.

**Table 2. Class distribution.**

|              | Oak-simple | Oak-intensive | Total |
|--------------|------------|---------------|-------|
| Wormhole     | 1638       | 2436          | 4074  |
| Crack        | 2086       | 2924          | 5010  |
| Training set | 2573       | 1949          | 4522  |
| Val set      | 1638       | 2436          | 4074  |
| Test set     | 1638       | 2436          | 4074  |

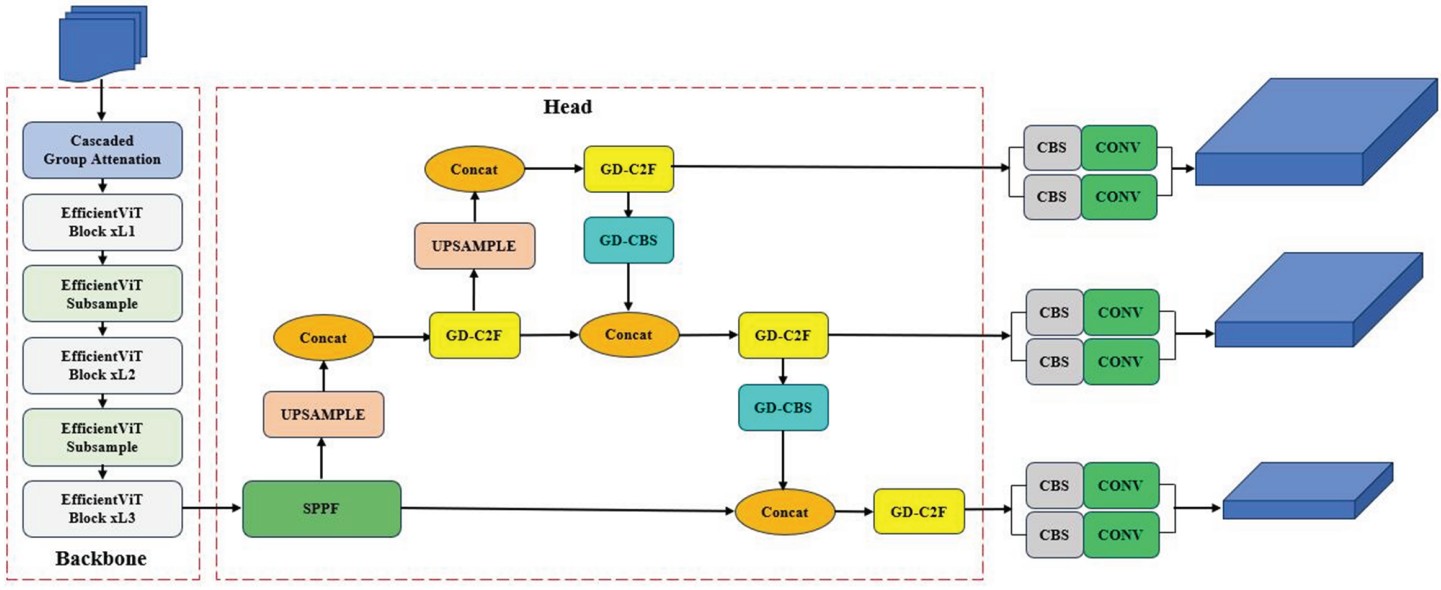

**Fig 4. The overall structure of the Oak-YOLO model.**

Shallow feature maps, in contrast to deep feature maps, have smaller receptive fields, higher spatial resolution, and more precise positional information, making them particularly effective for small object detection. This makes them highly suitable for the accurate detection of seed defects, such as cracks and insect holes. Therefore, to improve the detection performance for oak seed defects, this study integrates the Ghost module [24] with Dynamic Convolution [25] to construct a high-resolution Ghost-Dynamic prediction head, as illustrated in Fig 5. This approach effectively leverages the rich positional information and high resolution inherent in shallow features, enhancing the detection performance for small defect targets. Ghost-DynamicConv utilizes a two-step convolution operation by dividing the convolution process into primary and auxiliary convolutions. The primary convolution generates initial features $Y$, and the auxiliary convolution further optimizes the feature extraction. The final feature $F$ can be expressed as:

$$F = W_1 \cdot X + b_1 + f(W_2 \cdot X + b_2) \tag{4}$$

$W_1$ and $W_2$ represent the weights of the primary and auxiliary convolutions, respectively, and $f$ denotes the nonlinear activation function. The core of the dynamic weight allocation mechanism lies in selecting appropriate expert convolution kernels based on the features of the input image. This is achieved through a multi-layer perceptron (MLP), which generates dynamic weights. Specifically, the MLP takes the input image features as input and outputs a dynamic weight vector that indicates the activation level of different expert convolution kernels. The process of generating dynamic weights is as follows:

$$W_{\text{dynamic}} = W + \sigma(W_2 \cdot X + b_2) \tag{5}$$

As shown in Fig 6, To further enhance the richness of feature fusion, the PAN (Path Aggregation Network) structure is employed to fuse the $F_2$ feature map from the backbone network

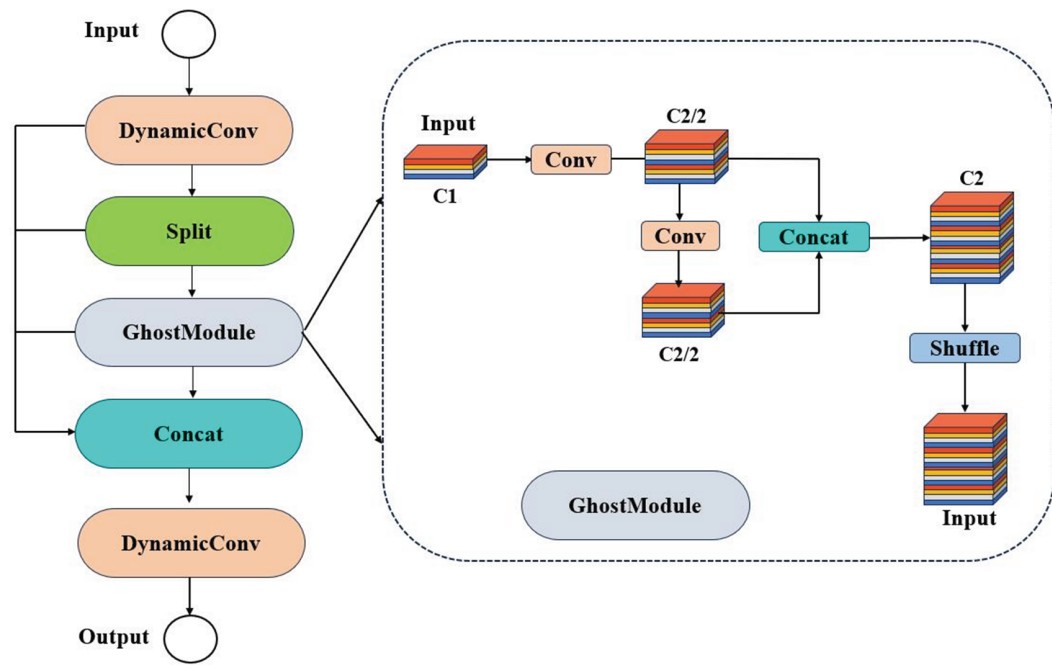

**Fig 5. Dynamic-Ghost architecture.**

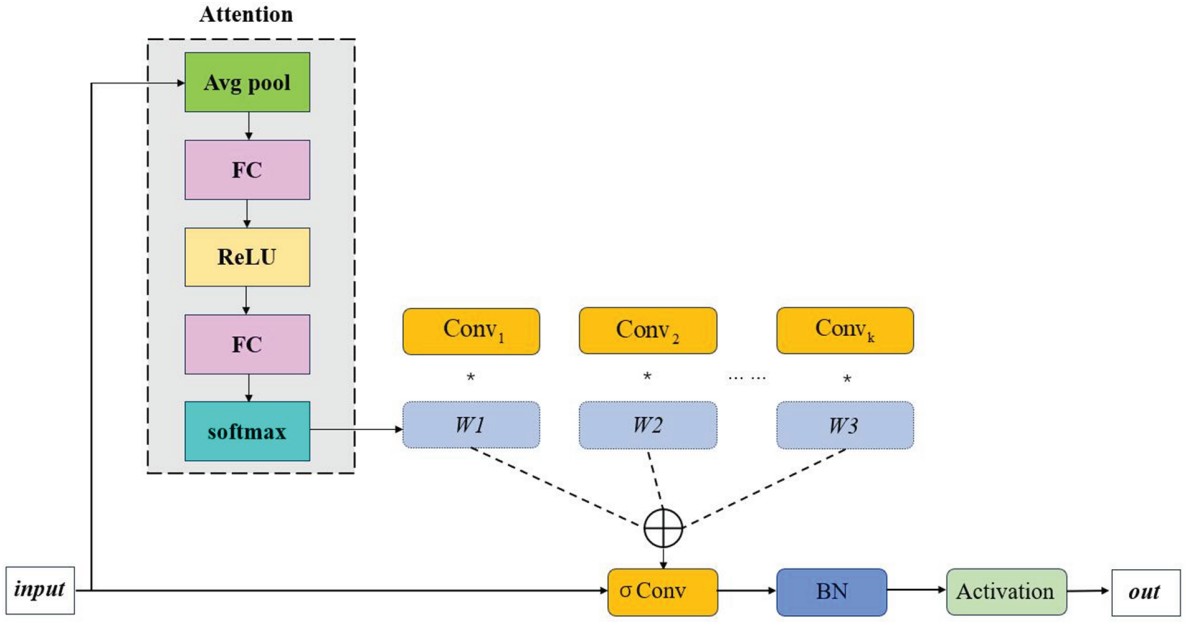

**Fig 6. Dynamic computation flow.**

with feature maps from other scales (e.g., $F_3$, $F_4$, etc.). This fusion process facilitates the creation of a smaller P2 prediction head. The introduction of the P2 prediction head provides additional positional and feature information for defect detection, while effectively reducing the loss of spatial features that can occur during down-sampling due to scale variations.

When combined with the other three prediction heads, the P2 head helps mitigate the decline in detection accuracy that is often caused by significant changes in object scales. This method enables the prediction heads to extract richer global features through a self-attention mechanism, which is particularly beneficial for the precise localization of defects in cases with significant overlap or occlusion.

**EfficientViT.** EfficientViT [26] is an efficient visual Transformer model designed to combine the strengths of Convolutional Neural Networks (CNN) and self-attention mechanisms, providing more precise and computationally efficient feature extraction for visual tasks. Fig 7 shows the structure of EfficientViT.In the context of oak seed defect detection, where defects such as cracks and insect holes are structurally complex, small in area, and irregular in shape, traditional CNN models face limitations in capturing long-range dependencies and global features. The core optimizations of EfficientViT are reflected in several key aspects. The "Sandwich Layout" effectively reduces memory consumption. Unlike conventional self-attention mechanisms, EfficientViT uses a single layer of self-attention for spatial feature mixing, supplemented by additional feed-forward network (FFN) layers before and after the

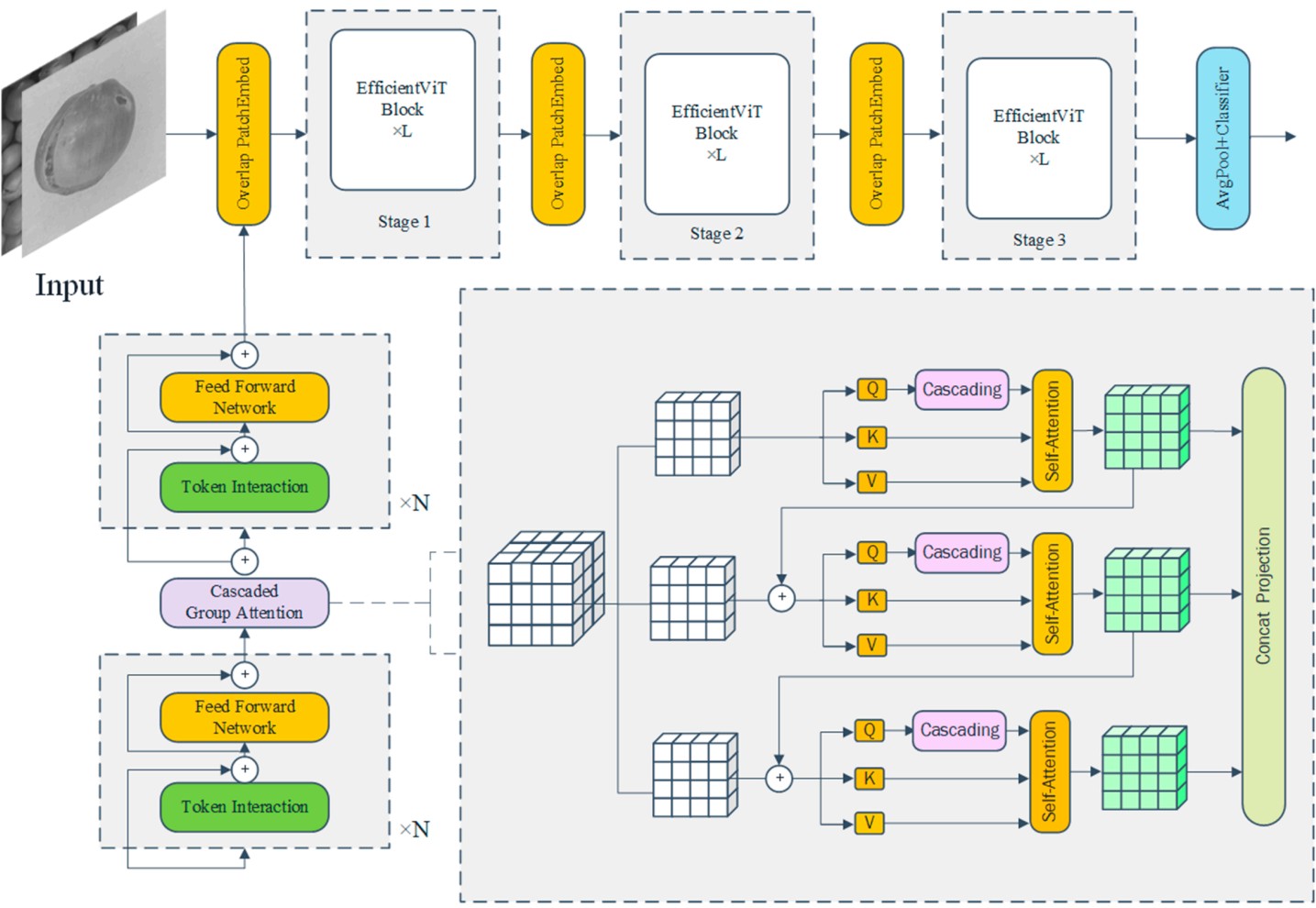

**Fig 7. EfficientViT architecture.**

self-attention layer to enhance communication between channels. The mathematical expression of this optimization is as follows:

$$X_{i+1} = \prod_i \varphi_i^F \left( \varphi_i^A \left( \prod_i \varphi_i^F (X_i) \right) \right) \tag{6}$$

$X_i$ represents the input features of the $i$-th layer, $\varphi_i^A$ denotes the self-attention layer, and $\varphi_i^F$ denotes the feed-forward network layer. By reducing the usage of self-attention layers, EfficientViT significantly lowers memory consumption while enhancing channel communication by increasing the number of feed-forward network layers. In the computation of Multi-Head Self-Attention (MHSA), EfficientViT introduces the Cascaded Group Attention (CGA) mechanism to further enhance computational efficiency. Unlike traditional MHSA models, CGA partitions the input features into multiple subsets, with each attention head processing only a portion of the features, thus reducing redundancy in feature computation. The core formula of this mechanism is given by:

$$\bar{X}_{ij} = \text{A\_ttn} \left( X_{ij} W_{ij}^Q, X_{ij} W_{ij}^K, X_{ij} W_{ij}^V \right) \tag{7}$$

$$\bar{X}_{i+1} = \text{Concat} \left( \bar{X}_{ij} \right)_{j=1}^h W_i^P \tag{8}$$

$\bar{X}_{ij}$ represents the self-attention result of the $j$-th head, $W_{ij}^Q$, $W_{ij}^K$, and $W_{ij}^V$ are the projection matrices, and $W_i^P$ is the final projection layer. Through this grouped computation, CGA effectively reduces the computational redundancy of MHSA, while the concatenation operation enhances the feature representational power.

**WIoUv3.** The size of the object is crucial in object detection tasks, especially in small object detection, where the standard IoU may not give sufficient attention due to the small area occupied by small objects. Existing bounding box loss functions, such as Smooth L1 [27], GIoU [28] and CIoU [29], often fail to effectively address the complex scenarios in defect detection, such as occlusion, small sizes, and class imbalance. To address this, WIoU introduces a weighted IoU calculation that assigns greater loss contributions to small objects. WIoU [30] is a weighted variant of the standard IoU, which adjusts the loss for each object or region by introducing a weight factor $w$ based on the standard IoU calculation. WIoUv3 improves upon this by introducing a dynamic non-monotonic focusing mechanism, allowing the model to allocate more computational resources to challenging samples with occluded or small objects, thus improving the accuracy and stability of the training process. Additionally, WIoUv3 provides a more refined method for bounding box regression by considering the overlap degree and dynamically adjusting the loss contribution of each bounding box. The weight factor based on the target center distance is defined as:

$$w_{\text{dist}} = \frac{1}{(d_{\text{center}} + \epsilon)^\beta} \tag{9}$$

where $\epsilon$ is a small constant to prevent division by zero, and $\beta$ is a hyperparameter that adjusts the impact of the distance. The formula for WIoU, incorporating the weight factor $w$, is expressed as:

$$\text{WIoU} = \frac{w \cdot (A \cap B)}{w \cdot (A \cup B)} = \frac{w_{\text{size}} \cdot w_{\text{dist}} \cdot (A \cap B)}{w_{\text{size}} \cdot w_{\text{dist}} \cdot (A \cup B)} \tag{10}$$

We conducted experiments on the dataset, comparing WIoUv3 with existing bounding box loss functions (such as Smooth L1, CIoU, and GIoU). As shown in Table 3. The experimental show that WIoUv3 significantly improves model performance, especially in handling complex scenarios like occlusion, small size, and class imbalance.

## Results and discussion

### Indicators for model evaluation

These metrics are used to evaluate the model's effectiveness in detecting seed cracks and insect hole defects, with the following formulas applied:

$$\text{Acc} = \frac{TP + TN}{TP + TN + FP + FN} \tag{11}$$

$$P = \frac{TP}{TP + FP} \tag{12}$$

$$R = \frac{TP}{TP + FN} \tag{13}$$

$$\text{AP} = \int_0^1 \text{Precision}(r)\,dr \tag{14}$$

$$\text{mAP} = \frac{1}{N}\sum_{r=1}^{N}\text{AP}_r \tag{15}$$

TP represents true positives (correct defect predictions), TN denotes true negatives (correct non-defect predictions), FP refers to false positives (incorrect defect predictions), FN indicates false negatives (missed defects), and N is the total number of categories. $R_r$ and $P_r$ represent recall and precision for the $r$-th class respectively.

### Experimental configuration

As shown in Table 4, the hardware and software environment of the experimental testing platform are listed below. We use the Adam optimizer to fine-tune the model parameters, Compared to traditional optimizers such as SGD, Adam provides adaptive learning rates and faster convergence, which is beneficial for models like Oak-YOLO that integrate complex modules such as EfficientViT and Ghost-DynamicConv. The training process consists of a maximum of 300 epochs, and the batch size is set to 16, with an initial learning rate set to 0.001.The hyperparameters used in training, including the learning rate, batch size, and

**Table 3. Performance comparison of different loss functions.**

| Loss Functions | Precision (%) | Recall (%) | mAP | F1-score (%) |
|---|---|---|---|---|
| Smooth L1 | 89.2 | 84.5 | 91.0 | 86.8 |
| CIoU | 90.1 | 85.3 | 92.2 | 87.6 |
| GIoU | 89.8 | 85.0 | 91.7 | 87.3 |
| WIoUv3 (Ours) | 94.5 | 87.8 | 94.5 | 90.1 |

**Table 4. Experimental test platform.**

| Hardware/Software | Model/Version | Details |
|---|---|---|
| GPU | NVIDIA GeForce RTX 3090Ti | Video Memory: 24 GB |
| Computer System | Windows 10 | RAM: 64 GB |
| Deep Learning Framework | PyTorch | Version: 1.12 |
| Computational Platform | CUDA | Version: 11.7 |
| IDE | PyCharm | Version: 2020.1.3 |
| Programming Language | Python | Version: 3.8 |

optimizer settings, were determined through manual tuning based on empirical performance on the validation set.

## Results of ablation experiments

To evaluate the effectiveness of each proposed module, a series of ablation experiments were conducted on the oak seed dataset using consistent training, validation, and testing protocols. The detailed quantitative results are summarized in Table 5. Integrating the Ghost-Dynamic module into the YOLOv8 detection head resulted in consistent improvements in mAP50, F1 score, and inference speed. Specifically, the enhanced YOLOv8-Ghost-Dynamic variant achieved an mAP50 of 95.97% and a precision of 98.61%, while also demonstrating a speed increase of 12.4 FPS (reducing inference time by 1.1 ms per image) compared to the baseline YOLOv8 model. Notably, the full Oak-YOLO configuration, which incorporates both the Ghost-Dynamic and EfficientViT modules, delivered the most substantial gains: it achieved an mAP50 of 96.92%, a precision of 98.12%, and an inference speed of 132.2 FPS (corresponding to 7.6 ms per image). These results underscore the effectiveness of the multi-module design in improving both accuracy and real-time performance.

As illustrated in Fig 8, precision and recall curves gradually converge after approximately 100 epochs, exhibiting minimal fluctuations in the later stages.Notably, Oak-YOLO consistently outperforms other models throughout training in both mAP0.5 and mAP0.5:0.95 metrics, achieving higher accuracy and faster convergence. This suggests that the integration of Ghost-Dynamic and EfficientViT modules enhances both the localization precision and the generalization capability of the model across different IoU thresholds.From the comparison of the confusion matrices, as shown in Fig 9. It is evident that YOLOv8 performs poorly in identifying cracks, with frequent missed detections. Following the enhancement, the issues of missed detections and misclassifications are significantly alleviated, achieving detection rates above 95% for both wormholes and cracks.

**Table 5. Ablation study results.**

| Ghost-Dynamic | EfficientViT | WIoUv3 | mAP50 | F1 (%) | Recall (%) | Parameter/M | Weight/MB | GFLOPS |
|---|---|---|---|---|---|---|---|---|
| | | | 90.5 | 88.8 | 86.5 | 37.2 | 74.8 | 12.6 |
| ✓ | | | 91.8 | 88.1 | 85.7 | 37.2 | 74.8 | 9.8 |
| | ✓ | | 93.2 | 90.8 | 86.2 | 37.2 | 64.8 | 7.3 |
| | | ✓ | 93.0 | 91.5 | 86.8 | 37.2 | 64.8 | 7.2 |
| ✓ | ✓ | | 93.7 | 92.4 | 87.6 | 36.0 | 68.2 | 7.0 |
| ✓ | | ✓ | 93.8 | 92.7 | 88.2 | 36.1 | 67.4 | 6.9 |
| | ✓ | ✓ | 94.0 | 94.10 | 89.6 | 35.6 | 66.0 | 6.8 |
| ✓ | ✓ | ✓ | 94.5 | 95.3 | 90.5 | 34.2 | 62.9 | 7.2 |

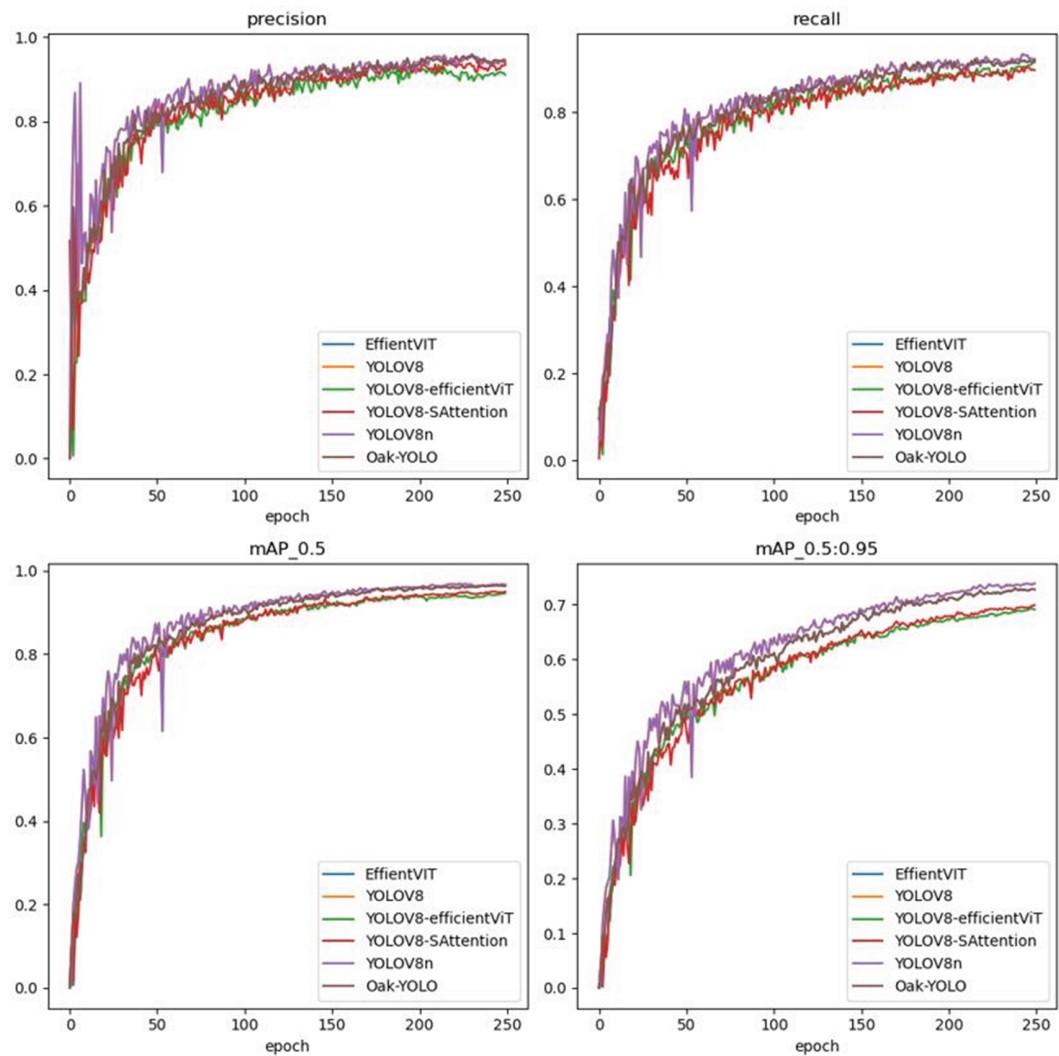

**Fig 8. Comparison of ablation experiments on different datasets.**

## Comparative performance analysis against alternative models

Based on the experimental results, our study primarily focuses on evaluating the differences between OSK-YOLO8 and YOLOv8 in terms of detection performance and inference speed. Two representative categories of object detection models were selected for comparative analysis: Transformer-based detectors, including rt-DETR-18 [31], rt-DETR-50 [31], and Deformable DETR [32]; and lightweight CNN-based detectors, comprising the YOLO series (YOLOv5s [33], YOLOv7 [34], YOLOv9 [35], YOLOv10 [36], YOLOv11 [37], and YOLOv12 [38]) as well as the proposed OSK-YOLO8.

As shown in Table 6, although Transformer-based models achieved superior detection accuracy, they require over 40 million parameters and entail high computational complexity. In contrast, the YOLO series strikes a better balance between accuracy and model efficiency. Notably, YOLOv5s and YOLOv7 have relatively small parameter sizes of only 16.4M and 18.8M, respectively. However, due to their limited number of convolutional layers and

## Normalized Confusion Matrix Comparison

**Fig 9. Comparison of Confusion Matrices between YOLOv8 and OSK-YOLO8.**

**Table 6. Comparative Evaluation of Detection Performance and Computational Efficiency Across Transformer-Based and YOLO Series Models.**

| Models | mAP50 (%) | FPS | Parameters (M) | GFLOPS |
|---|---|---|---|---|
| rt-DETR-18 | 96.4 | 75 | 39.5 | 56.9 |
| rt-DETR-50 | 98.1 | 60 | 48.2 | 92.4 |
| Deformable DETR | 97.8 | 55 | 48.7 | 78.6 |
| YOLOv5s | 92.1 | 230 | 16.4 | 8.5 |
| YOLOv7 | 93.4 | 211 | 18.8 | 9.4 |
| YOLOv9 | 94.5 | 195 | 21.2 | 12.3 |
| YOLOv10 | 95.7 | 162 | 24.5 | 18.7 |
| YOLOv11 | 96.8 | 135 | 27.1 | 22.4 |
| YOLOv12 | 97.3 | 120 | 29.8 | 29.2 |
| Ours (Oak-YOLO) | 96.2 | 264 | 34.2 | 7.2 |

channels, these models struggle to extract effective features in complex backgrounds. It is worth highlighting that YOLOv9 to YOLOv12 exhibit a consistent improvement in detection performance while maintaining the efficiency of the YOLO architecture. Specifically, YOLOv9 achieves an mAP50 of 94.5%, which increases to 97.3% in YOLOv12. Nonetheless, this performance gain comes at the cost of increased model size and complexity—parameter counts grow from 21.2M to 29.8M, GFLOPs rise from 12.3 to 29.2, and inference speed declines from 195 FPS to 120 FPS, indicating a trade-off between performance and computational cost.

Among all evaluated models, OSK-YOLO8 offers the most favorable trade-off between performance and efficiency. It achieves an mAP50 of 96.2%, ranking among the top within the YOLO family. With only 12.0M parameters and 6.4 GFLOPs, it reaches an inference speed of 264 FPS—substantially outperforming all baseline models, including YOLOv12. Compared with YOLOv12, OSK-YOLO8 reduces GFLOPs by approximately 78%, parameter count by nearly 60%, and improves inference speed by over 120 FPS. Fig 10 provides a visual

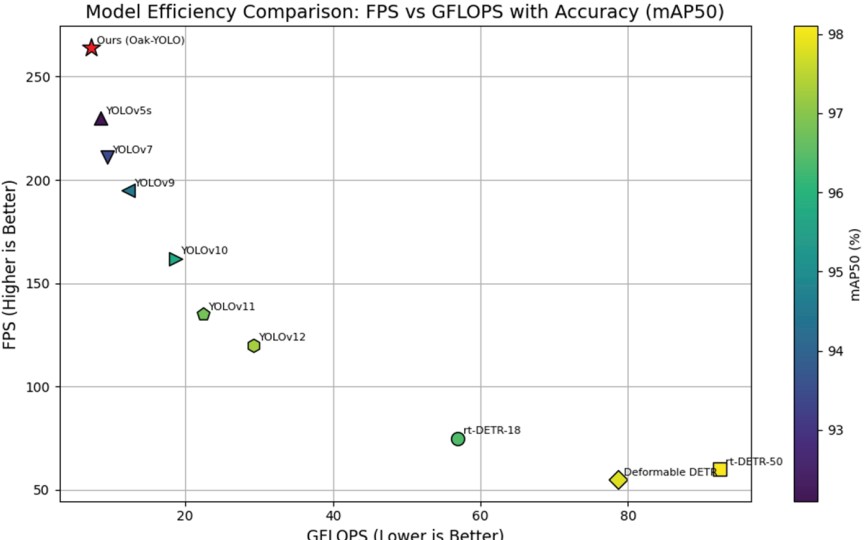

**Fig 10. Comparative Evaluation of Detection Performance and Computational Efficiency Across Transformer-Based and YOLO Series Models.**

comparison of each model's accuracy, speed, and computational complexity, clearly demonstrating the comprehensive advantages of OSK-YOLO8.

## Verify the results of the experiment

To evaluate the effectiveness of OSK-YOLO8 and YOLOV8n in detecting defects during actual production, this study applied the pre-trained OSK-YOLO8 and YOLOV8n models to validation experiments involving seed images. Fig 11 visually represents the detection results of OSK-YOLO8.

The visualization results demonstrate that in the Oak-Intensive task, characterized by densely packed seeds, the YOLOV8n model struggles to accurately identify defects, especially

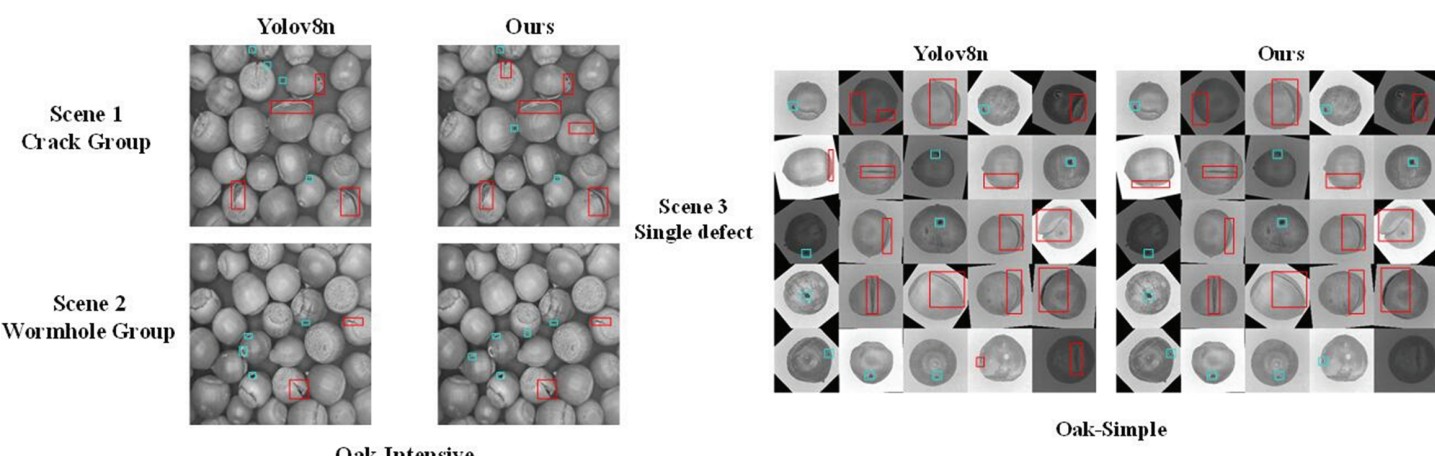

**Fig 11. Experimental results verification, illustrating the comparison between predicted and actual outcomes.**

**Table 7. Cross-device performance comparison.**

| Test Set | mAP50 (%) | mAP50–95 (%) | Precision (%) | Recall (%) | F1 Score (%) |
|---|---|---|---|---|---|
| Test1 (Oak-simple) | 96.1 | 76.3 | 95.3 | 91.6 | 93.4 |
| Test2 (Oak-intensive) | 95.8 | 75.1 | 94.9 | 90.7 | 92.8 |
| Test3 (OnePlus) | 94.7 | 70.2 | 92.4 | 88.3 | 90.3 |
| Test4 (iPhone) | 93.8 | 68.9 | 92.1 | 87.5 | 89.7 |

in scale scenarios lacking fine detail, particularly when cracks dominate the scene. In contrast, OSK-YOLO8, with its dynamic detection head, significantly enhances the expression of fine-grained details.

Similarly, in scenarios dominated by wormholes, YOLOV8n often misjudges fine details, whereas OSK-YOLO8 almost accurately identifies all wormhole cases. In single-defect scenarios, YOLOV8n tends to misinterpret the seed base as a crack defect, while OSK-YOLO8 exhibits outstanding performance in crack detection. Both models demonstrate similar performance in wormhole identification, though YOLOV8n occasionally misclassifies surface textures as wormholes, an issue not observed with OSK-YOLO8.

## Robustness evaluation

To validate the model's robustness across devices and scenarios, we conducted cross-domain testing using images captured by mobile devices. The external validation sets included: 1) OnePlus ACE2 Pro (OnePlus Technology Co., Ltd., Shenzhen, China) and 2) iPhone 13 Plus (Apple Inc., USA), with resolutions of 3,264×2,448 and 3,024×4,032 respectively.

As shown in Table 7, Oak-YOLO achieved mAP50 scores of 94.7% and 93.8% on mobile-captured test sets (Test3 and Test4), reflecting only a 2.3–3.3% drop compared to the laboratory-controlled environments (Test1 and Test2). Despite the more challenging conditions associated with mobile devices—such as inconsistent lighting and complex backgrounds—the model maintained a high precision of over 92% and F1 scores above 89%, indicating stable and reliable detection performance. Furthermore, the mAP50–95 values on mobile-captured sets remained above 68%, further demonstrating the model's generalization capability across diverse imaging sources.

To evaluate the defect localization capability of the model in cross-domain scenarios, this study uses the Grad-CAM++ heatmap visualization method [39] to systematically compare the attention region distributions of Oak-YOLO and YOLOv8 under complex backgrounds. The heatmaps use a blue-to-red color spectrum (low-to-high values) to visually present the response intensity of the model to the input images. As shown in Fig 12, Oak-YOLO exhibits more concentrated high-response regions in images captured by mobile devices, with activation area coverage exceeding 85%, accurately focusing on seed defect locations. In contrast, YOLOv8 shows more dispersed attention distribution, with 12–15% false activations near seed edges and greater susceptibility to background noise.

## Conclusion

In this study, we proposed Oak-YOLO, an improved YOLOv8-based detection framework designed specifically for identifying defects in oak seeds. The model integrates the EfficientViT backbone for enhanced global feature extraction, and introduces a Ghost-Dynamic prediction head to better detect small and irregular targets. Furthermore, the adoption of the WIoUv3 loss function improves bounding box regression for overlapping and deformable defects such as cracks and insect holes.

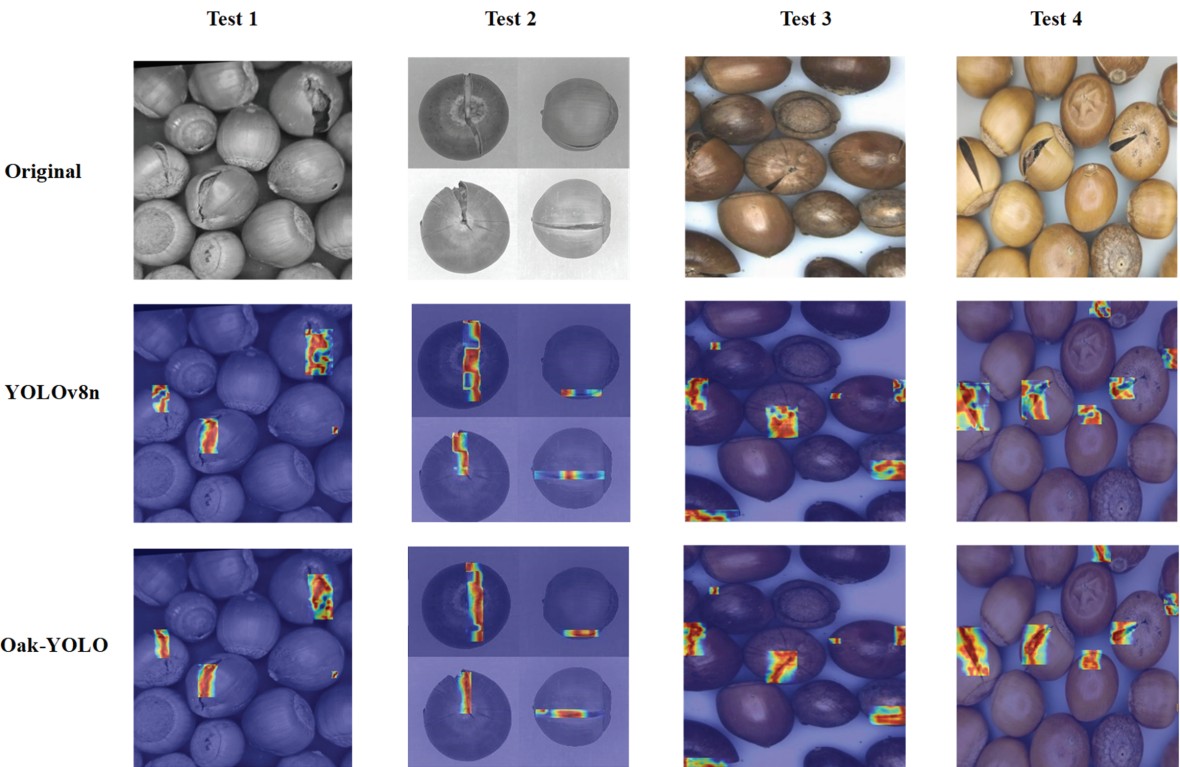

**Fig 12. Attention heatmap comparison between Oak-YOLO and YOLOv8.**

Experimental evaluations demonstrated that Oak-YOLO achieved a mAP50 of 94.5%, an F1-score of 95.3%, and a detection speed of 132.2 FPS on the oak-intensive dataset, significantly outperforming the baseline YOLOv8 model. In cross-device validation using smartphone-captured images, the model maintained high accuracy and robustness, confirming its generalization ability across diverse acquisition environments. Comparative analysis also showed that Oak-YOLO offers superior performance-efficiency trade-offs compared to state-of-the-art YOLO variants and Transformer-based models.

These findings highlight the practical applicability of Oak-YOLO for real-time and high-precision seed defect detection in forestry. Future work will focus on further reducing computational costs and extending the model to support defect screening across a broader range of tree species.

## Author contributions

**Conceptualization:** Zhuqi Li, Wangyu Wu.

**Data curation:** Wangyu Wu.

**Funding acquisition:** Hongbo Mu.

**Methodology:** Dongkui Chen.

**Writing – original draft:** Hao Li.

**Writing – review & editing:** Zhuqi Li, Xuanlong He, Hongbo Mu.

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
