## [Decision Letter · Decision Letter 0]

2 May 2025

PONE-D-25-18128Oak-YOLO: A High-Performance Detection Model for Automated Oak Seed Defect IdentificationPLOS ONE

Dear Dr. Mu,

Thank you for submitting your manuscript to PLOS ONE. After careful consideration, we feel that it has merit but does not fully meet PLOS ONE’s publication criteria as it currently stands. Therefore, we invite you to submit a revised version of the manuscript that addresses the points raised during the review process.

Please revise the paper according to the reviewer comments.

We look forward to receiving your revised manuscript.

Kind regards,

Fatih Uysal, Ph.D.

Academic Editor

PLOS ONE

2. Please update your submission to use the PLOS LaTeX template. The template and more information on our requirements for LaTeX submissions can be found at http://journals.plos.org/plosone/s/latex

3. Please ensure you have stated the source of the oak seeds used in the study.

5. Please note that PLOS ONE has specific guidelines on code sharing for submissions in which author-generated code underpins the findings in the manuscript. In these cases, we expect all author-generated code to be made available without restrictions upon publication of the work. Please review our guidelines at https://journals.plos.org/plosone/s/materials-and-software-sharing#loc-sharing-code and ensure that your code is shared in a way that follows best practice and facilitates reproducibility and reuse.

6. Thank you for stating the following financial disclosure:

“the Fundamental Research Funds for the Central Universities-No. 2572023DJ02”

7. When completing the data availability statement of the submission form, you indicated that you will make your data available on acceptance. We strongly recommend all authors decide on a data sharing plan before acceptance, as the process can be lengthy and hold up publication timelines. Please note that, though access restrictions are acceptable now, your entire data will need to be made freely accessible if your manuscript is accepted for publication. This policy applies to all data except where public deposition would breach compliance with the protocol approved by your research ethics board. If you are unable to adhere to our open data policy, please kindly revise your statement to explain your reasoning and we will seek the editor's input on an exemption. Please be assured that, once you have provided your new statement, the assessment of your exemption will not hold up the peer review process.

8. Please amend your authorship list in your manuscript file to include author Wangyu Wu.

Additional Editor Comments:

Please revise the paper according to the reviewer comments.

Reviewers' comments:

Reviewer's Responses to Questions

**Comments to the Author**

1. Is the manuscript technically sound, and do the data support the conclusions?

Reviewer #1: Yes

Reviewer #2: Partly

2. Has the statistical analysis been performed appropriately and rigorously? 

Reviewer #1: Yes

Reviewer #2: No

3. Have the authors made all data underlying the findings in their manuscript fully available?

Reviewer #1: No

Reviewer #2: Yes

4. Is the manuscript presented in an intelligible fashion and written in standard English?

Reviewer #1: No

Reviewer #2: Yes

5. Review Comments to the Author

Reviewer #1: Dear Autor(s):

The manuscript presents a promising approach for oak seed defect detection using an enhanced YOLO-based deep learning model. While the study is technically sound and addresses a relevant problem in forestry and agricultural automation, several aspects need to be clarified, revised, or improved to meet the standards of a high-impact journal.

1. Revise the reference list to comply with the journal's formatting guidelines. For example, the first cited reference appears as number 15 (line 33), which suggests inconsistency in numbering or missing earlier references.

2. Lines 215–217 state that YOLOv8 is the latest version of the YOLO series, which is currently incorrect. The most recent version is YOLOv12. While YOLOv8 may have been the latest at the time of the study, it is recommended that the authors also evaluate their method against more recent versions for a more robust comparison.

3. Clarify the rationale behind using the Adam optimization algorithm. Was the choice of hyperparameters based on manual tuning or derived through a specific algorithmic approach?

4. Including one or more state-of-the-art YOLO models (e.g., YOLOv9–YOLOv12) in Table 4 would provide a valuable benchmark and allow for a better evaluation of Oak-YOLO’s performance against current models.

5. The dataset is reported to be split in an 8:1:1 ratio, but it is unclear what operations were performed using the test set. Moreover, the test results are not explicitly presented. It should also be clarified whether class balance was considered during dataset splitting (e.g., through stratified random sampling).

6. Is the dataset openly accessible? The manuscript states that it can be obtained by contacting the corresponding author, but according to PLOS ONE’s data availability policy, this may not be sufficient. Authors should consider uploading the dataset to a public repository with a DOI.

7. The technical writing should be simplified for clarity and consistency. In particular, the abstract should be rewritten using concise and impactful language to better communicate the novelty and significance of the work.

I believe that with the suggested revisions and clarifications, this manuscript can make a valuable contribution to the field and meet the publication standards of the journal.

Reviewer #2: The aim of the paper is to develop an advanced, efficient, and accurate object detection model—Oak-YOLO—tailored specifically for identifying defects in oak seeds, such as insect holes and cracks, which are critical for seed quality and germination. By enhancing the YOLOv8 architecture with components like EfficientViT, Ghost-DynamicConv, and the WIoUv3 loss function, the proposed model achieves superior performance. Experimental results demonstrate a high detection accuracy of 94.3% and a mAP50 of 96.2%, with significantly improved speed and precision compared to baseline models, making it well-suited for real-time industrial applications in forestry.

Comments to the Authors:

The novelty of Oak-YOLO is not sufficiently distinguished from existing YOLO-based models; clearer justification is needed.

The dataset used is limited in size and variability; generalizability is questionable.

There is a lack of comparison with non-YOLO architectures beyond brief mentions—benchmarking against more diverse baselines is necessary.

The ablation study could be expanded to isolate the impact of each individual module.

Details on statistical significance and error analysis are missing.

6. PLOS authors have the option to publish the peer review history of their article (what does this mean?). If published, this will include your full peer review and any attached files.

Reviewer #1: No

Reviewer #2: No

---

## [Author Response · Author response to Decision Letter 1]

22 May 2025

Response to Reviewers

Manuscript Number: PONE-D-25-18128

Manuscript Title: Oak-YOLO: A High-Performance Detection Model for Automated Oak Seed Defect Identification

Dear Editors and Reviewers,

We would like to express our sincere gratitude to the Editors and Reviewers for your valuable time, insightful feedback, and constructive suggestions regarding our manuscript entitled "Oak-YOLO: improved YOLOV8 with multi-scale feature fusion in the detection of defects in oak seeds". We are deeply appreciative of your thoughtful comments, which have greatly contributed to improving the clarity, rigor, and overall quality of our work.

We have carefully addressed each point raised by the reviewers and revised the manuscript accordingly. Please find below our detailed point-by-point responses. All modifications in the manuscript are clearly marked for your convenience.

Hongbo Mu

(On behalf of all authors)

Corresponding author

Email: mhb506@nefu.edu.cn

Responses to Editorial Comments

We sincerely thank the Editorial Office for the detailed instructions and helpful reminders. We have carefully followed each point and updated the manuscript and submission materials accordingly. Please find our point-by-point responses below:

Response: We have renamed our files according to the PLOS ONE guidelines and ensured that the manuscript fully complies with the journal’s style requirements, following the templates provided.

2. Please update your submission to use the PLOS LaTeX template.

Response: Thank you for the guidance. We have fully migrated our manuscript into the official PLOS LaTeX template, and have carefully checked the formatting to ensure consistency with journal standards.

3. Please ensure you have stated the source of the oak seeds used in the study.

Response: We have added a clear statement of the oak seed source in the “Materials and Methods” section:

“The oak seeds used in this study were purchased from local oak farmers in Xinxu Town, Suqian, Jiangsu Province, China.”

5. Please review our code sharing policy and ensure your code is shared appropriately.

Response: We fully support the journal’s policy on code transparency and reproducibility. All author-generated code has been uploaded Figshare and is available at: https://doi.org/10.6084/m9.figshare.29072015.

6. Please state what role the funders took in the study.

Response: We sincerely thank the editor for this important reminder. As the funders provided assistance in the review and editing of the manuscript, we have amended the funding statement accordingly. The updated funder role statement now reads:

“The funders provided support in reviewing and editing the manuscript.”

This has been included both in the manuscript and in the cover letter, as requested.

7. Data availability policy.

Response: We have now made our dataset publicly available through Figshare. The DOI is: https://doi.org/10.6084/m9.figshare.29072015

This DOI has been included in the revised Data Availability Statement and in the main text of the manuscript. We confirm that the data are now fully open access and no restrictions apply.

8. Please amend your authorship list to include author Wangyu Wu.

Response: We sincerely apologize for the previous omission. We have now added author Wangyu Wu to the author list in the manuscript file, and appropriately updated the Author Contributions section.

Reviewer #1

Comment 1: Revise the reference list to comply with the journal's formatting guidelines. For example, the first cited reference appears as number 15 (line 33), which suggests inconsistency in numbering or missing earlier references.

Response: Thank you very much for pointing this out. We have thoroughly reviewed and corrected the reference numbering throughout the manuscript to ensure that all citations appear in sequential order in accordance with the journal's formatting requirements.

Comment 2: Lines 215–217 state that YOLOv8 is the latest version of the YOLO series, which is currently incorrect. The most recent version is YOLOv12. While YOLOv8 may have been the latest at the time of the study, it is recommended that the authors also evaluate their method against more recent versions for a more robust comparison.

Response: We sincerely thank the reviewer for highlighting this issue. We have clarify that YOLOv8 was the latest version at the time our study was initiated. In response to your suggestion, we have extended our experimental comparison to include YOLOv9 to YOLOv12. The updated results are presented in Table 6 and discussed in the revised manuscript (Section: Comparative performance analysis against alternative models).

Comment 3: Clarify the rationale behind using the Adam optimization algorithm. Was the choice of hyperparameters based on manual tuning or derived through a specific algorithmic approach?

Response: Thank you for your helpful suggestion. We have revised the “Experimental Configuration” section to explicitly state that the Adam optimizer was selected due to its adaptive learning rate and faster convergence, particularly suitable for our network structure. The hyperparameters were chosen through manual tuning based on empirical performance on the validation set.

Comment 4: Including one or more state-of-the-art YOLO models (e.g., YOLOv9–YOLOv12) in Table 4 would provide a valuable benchmark and allow for a better evaluation of Oak-YOLO’s performance against current models.

Response: We fully agree with your suggestion. We have incorporated the performance metrics of YOLOv9 through YOLOv12 into Table 6 , and provided a detailed comparative analysis in the corresponding section. This addition allows for a more comprehensive and up-to-date evaluation of our proposed model.

Comment 5: The dataset is reported to be split in an 8:1:1 ratio, but it is unclear what operations were performed using the test set. Moreover, the test results are not explicitly presented. It should also be clarified whether class balance was considered during dataset splitting (e.g., through stratified random sampling).

Response: Thank you for raising this important point. We have revised the “Data Augmentation” section to clarify that the test set was reserved exclusively for final evaluation and was not involved in any model training or tuning. Additionally, we employed stratified random sampling during data splitting to maintain class balance across training, validation, and testing sets. The test results are now explicitly presented in Table 7 .

Comment 6: Is the dataset openly accessible? The manuscript states that it can be obtained by contacting the corresponding author, but according to PLOS ONE’s data availability policy, this may not be sufficient.

Response: Thank you for bringing this to our attention. In compliance with the PLOS ONE data availability policy, we have uploaded our dataset to Figshare, where it is now publicly accessible via the following DOI: https://doi.org/10.6084/m9.figshare.29072015.

Comment 7: The technical writing should be simplified for clarity and consistency. In particular, the abstract should be rewritten using concise and impactful language to better communicate the novelty and significance of the work.

Response: We appreciate this thoughtful suggestion. We have revised the abstract extensively to enhance its clarity, conciseness, and emphasis on the novelty of our proposed framework. We have also reviewed and refined the manuscript throughout to improve overall readability and consistency.

Reviewer #2

Comment 1: The novelty of Oak-YOLO is not sufficiently distinguished from existing YOLO-based models; clearer justification is needed.

Response: Thank you for this valuable feedback. We have revised the Introduction and Conclusion sections to more clearly articulate the novel contributions of Oak-YOLO. Specifically, our model introduces a high-resolution Ghost-Dynamic head, an EfficientViT-enhanced backbone, and the WIoUv3 loss to better detect small, irregular, and overlapping defects—improvements not found in previous YOLO variants.

Comment 2: The dataset used is limited in size and variability; generalizability is questionable.

Response: We acknowledge this concern and have taken steps to address it. In the revised manuscript, we conducted additional cross-device generalization experiments using oak seed images captured with mobile devices (OnePlus ACE2 Pro and iPhone 13 Plus). The results demonstrate that Oak-YOLO maintains high detection accuracy and robustness across different image acquisition platforms.

Comment 3: There is a lack of comparison with non-YOLO architectures beyond brief mentions—benchmarking against more diverse baselines is necessary.

Response: We appreciate the reviewer’s insight. We have extended the scope of our comparative study to include several Transformer-based detectors (e.g., rt-DETR-18, rt-DETR-50, Deformable DETR). The updated comparison in Table 6 allows for a broader assessment of our model’s performance relative to both CNN- and Transformer-based architectures.

Comment 4: The ablation study could be expanded to isolate the impact of each individual module.

Response: Thank you for the suggestion. We have expanded the ablation study to comprehensively evaluate the individual and combined effects of Ghost-DynamicConv, EfficientViT, and WIoUv3. The new results are presented in Table 5 and Fig 8,9 illustrating the performance gain brought by each component.

Comment 5: Details on statistical significance and error analysis are missing.

Response: We agree with the importance of including statistical evidence. We included confusion matrix visualizations and Grad-CAM++ heatmaps to provide qualitative analysis and interpretability of model behavior.

Once again, we are deeply grateful for your invaluable comments and suggestions, which have significantly improved the quality of our manuscript. We respectfully hope that the revised version now meets the high standards required for publication in PLOS ONE.

We thank the Editors and Reviewers again for their insightful feedback, which has helped improve our manuscript significantly. We hope the revised version meets the expectations for publication in PLOS ONE.

---

## [Decision Letter · Decision Letter 1]

15 Jun 2025

Oak-YOLO: A High-Performance Detection Model for Automated Oak Seed Defect Identification

PONE-D-25-18128R1

Dear Dr. Mu,

We’re pleased to inform you that your manuscript has been judged scientifically suitable for publication and will be formally accepted for publication once it meets all outstanding technical requirements.

Kind regards,

Fatih Uysal, Ph.D.

Academic Editor

PLOS ONE

Additional Editor Comments (optional):

Considering the current quality of the study, the authors' responses to the reviewers' comments, and the reviewers' final recommendations, the paper has been accepted as it demonstrates strong potential to contribute to the literature.

Reviewers' comments:

Reviewer's Responses to Questions

**Comments to the Author**

1. If the authors have adequately addressed your comments raised in a previous round of review and you feel that this manuscript is now acceptable for publication, you may indicate that here to bypass the “Comments to the Author” section, enter your conflict of interest statement in the “Confidential to Editor” section, and submit your "Accept" recommendation.

Reviewer #1: All comments have been addressed

Reviewer #2: All comments have been addressed

2. Is the manuscript technically sound, and do the data support the conclusions?

Reviewer #1: Yes

Reviewer #2: Yes

3. Has the statistical analysis been performed appropriately and rigorously? 

Reviewer #1: Yes

Reviewer #2: Yes

4. Have the authors made all data underlying the findings in their manuscript fully available?

Reviewer #1: Yes

Reviewer #2: Yes

5. Is the manuscript presented in an intelligible fashion and written in standard English?

Reviewer #1: Yes

Reviewer #2: Yes

6. Review Comments to the Author

Reviewer #1: The manuscript has been successfully revised according to reviewer comments and journal policies. It meets PLOS ONE's publication criteria for scientific contribution, methodological soundness, data accessibility and transparency.

Reviewer #2: (No Response)

7. PLOS authors have the option to publish the peer review history of their article (what does this mean?). If published, this will include your full peer review and any attached files.

Reviewer #1: No

Reviewer #2: No

---

## [Editor Report · Acceptance letter]

PONE-D-25-18128R1

PLOS ONE

Dear Dr. Mu,

I'm pleased to inform you that your manuscript has been deemed suitable for publication in PLOS ONE. Congratulations! Your manuscript is now being handed over to our production team.

Kind regards,

on behalf of

Assoc. Prof. Dr. Fatih Uysal

Academic Editor

PLOS ONE